# CD8 and CD4 T Cell Populations in Human Kidneys

**DOI:** 10.3390/cells10020288

**Published:** 2021-02-01

**Authors:** Carlos van der Putten, Ester B.M. Remmerswaal, Matty L. Terpstra, Nelly D. van der Bom, Jesper Kers, Ineke J.M. ten Berge, Suzanne E. Geerlings, René A.W. van Lier, Frederike J. Bemelman, Michiel C. van Aalderen

**Affiliations:** 1Department of Experimental Immunology, Amsterdam institute for Infection & Immunity, Amsterdam UMC, University of Amsterdam, 1105AZ Amsterdam, The Netherlands; e.b.remmerswaal@amsterdamumc.nl (E.B.M.R.); Matty.Terpstra@amsterdamumc.nl (M.L.T.); n.d.baylon@amsterdamumc.nl (N.D.v.d.B.); r.j.tenberge@amsterdamumc.nl (I.J.M.t.B.); f.j.bemelman@amsterdamumc.nl (F.J.B.); M.C.vanaalderen@amsterdamumc.nl (M.C.v.A.); 2Division of Internal Medicine, Department of Nephrology, Renal Transplant Unit, Amsterdam Infection & Immunity Institute (AI&II), Amsterdam UMC, University of Amsterdam, 1105AZ Amsterdam, The Netherlands; 3Department of Pathology, Leiden University Medical Center, 2300 RC Leiden, The Netherlands; j.kers@amsterdamumc.nl; 4Department of Internal Medicine, Infectious Diseases, Amsterdam Infection & Immunity Institute (AI&II), Amsterdam UMC, University of Amsterdam, 1105AZ Amsterdam, The Netherlands; s.e.geerlings@amsterdamumc.nl; 5Sanquin Research and Landsteiner laboratory, 1066CX Amsterdam, The Netherlands; r.vanlier@amsterdamumc.nl

**Keywords:** tissue-resident lymphocytes, T-cells, CD8, CD4, CD69, CD103, kidney, allograft

## Abstract

Background: At border sites, and in internal organs, tissue resident memory T cells (T_RM)_ contribute to the immune barrier against pathogens like viruses, bacteria, fungi, and cancer. However, information on the presence and function of these cells in the human kidney is scant. In order to better understand the T cell-mediated immunological defense in this organ, we aimed to determine phenotypic and functional aspects of CD8 and CD4 T cells present in healthy and allograft kidney tissue. Methods: Using multichannel flow cytometry, we assessed the phenotype and function of T cells in healthy renal tissue samples (*n* = 5) and kidney allograft tissue (*n* = 7) and compared these aspects to T cells in peripheral blood from healthy controls (*n* = 13). Results: Kidney tissue samples contained substantial amounts of CD8 and CD4 T cells. In contrast to the circulating cells, kidney T cells frequently expressed CD69 and CD103, and were more often actively cycling. Furthermore, nearly all kidney T cells expressed CXCR3, and often expressed CXCR6 compared to T cells in the circulation. Markedly, kidney T cells produced greater quantities of IFNγ than circulating cells and were frequently polyfunctional. Conclusion: Functional T cells with the characteristic traits of T_RM_ reside in human kidney tissues. These cells are more often actively cycling and frequently express CXCR3 and CXCR6.

## 1. Introduction

Resident memory T cells (T_RM_) persist in tissues to provide a long-term localized defense against pathogens. In contrast to recirculating T cells, T_RM_ populations cannot be detected in peripheral blood, and differ significantly from recirculating T cell populations with regard to their phenotype, function, and metabolism [1]. Indeed, this is not surprising considering the fact that different organ systems, like the lungs, are exposed to higher loads of different pathogens than is the case for the circulation, which is generally isolated from the outside world [2].

Tissue retention of T_RM_ persists for years and is mediated by mechanisms that involve the Sphingosine-1-phosphate receptor 1 (S1PR1)-sphingosine 1-phosphate (SP1) axis. S1PR1 is a receptor expressed by T cells that mediates egression from secondary lymphoid organs upon binding to the bioactive signaling molecule SP1, which is an important mediator of T cell trafficking. This axis can be interrupted by CD69, a membrane-bound c-type lectin that antagonizes S1PR1 expression, thus mediating retention. Furthermore, in mice S1PR1 expression is induced by the transcription factor Krüppel-like Factor 2 (KLF2), which was found to be downregulated in T_RM_, thereby also inhibiting lymphocyte egress [3]. In turn, downregulation of KLF2 was shown to be mediated by the transcription factor homolog of Blimp1 in T cells (Hobit), which is expressed in a T_RM_-specific manner in murine T cells [4]. Furthermore, some T_RM_ also express CD103 (integrin αE), whereby T_RM_ can be identified using CD69 and CD103 expression [1].

While knowledge on T_RM_ phenotype and function is increasing for various human tissues, little is known about these aspects in the kidney. Here, T cells are exposed to a distinct set of pathogens such as bacteria ascending from the lower urinary tract and renotropic viruses like polyomavirus BK (BKPyV). Indeed, we recently described how human kidneys contain a subset of T cells expressing CD69 and/or CD103, amongst which there are CD8 T cells specifically targeting the BKPyV proteins virion protein 1 (VP1) and large T antigen (LTAG) protein [5]. We, and others, also demonstrated the presence of CD69/CD103-expressing mucosa-associated invariant T (MAIT) cells in kidney tissue [6,7].

To better understand what kind of T cells constitute the localized immune response in this organ, we used multichannel flowcytometry to investigate the phenotype and function of kidney T_RM_s, isolated from donor kidney material of renal transplant recipients (RTRs) and from healthy kidney tissue adjacent to renal clear cell carcinomas, to see how these populations compare to T cell populations in the circulation.

We found that human kidney tissue holds substantial amounts of CD4 and CD8 T cells expressing just CD69, CD69 and CD103, or none of these markers. We found that CD69^−^CD103^−^ cells in kidney tissue differ substantially from T cells in the circulation and may represent a T_RM_ population lacking the canonical T_RM_ markers. Furthermore, we show that kidney T cells are more often actively cycling, as judged by increased expression of Ki-67, when compared to blood. We also found that CD8 and CD4 T cells in kidney tissue nearly always express CXCR3 and often express CXCR6. These findings implicate a role for these chemokine receptors in the attraction and maintenance of kidney-resident memory T cell populations.

## 2. Materials and Methods

### 2.1. Patients and Samples

Samples were obtained from the Biobank Renal Diseases of the Amsterdam UMC location AMC. In this Biobank, patient samples, such as blood and kidney tissue, are collected and stored from healthy living kidney donors and RTRs that are followed before and after kidney transplantation. This study was conducted according to the principles outlined in the Declarations of Helsinki and Istanbul and all participants provided written informed consent prior to enrollment in the Biobank. Additionally, residual tissue from patients who underwent tumor nephrectomy (kidney tissue distant from the tumor) was donated by the Department of Pathology and also stored in the Biobank. These tissues were processed anonymously according to the Federation of Dutch Medical Scientific Societies’ Code of Conduct (Human Tissue and Medical Research: Code of Conduct for Responsible Use, 2011 www.federa.org).

### 2.2. Peripheral Blood Mononuclear Cells (PBMCs)

Blood samples were obtained from cytomegalovirus (CMV)-seronegative healthy controls (living kidney donors prior to surgery, *n* = 13). Characteristics of the participants in this study are displayed in Table 1. PBMCs were isolated from sodium heparin blood by standard density gradient centrifugation and subsequently cryopreserved until the day of analysis.

### 2.3. Kidney Tissue

Healthy kidney tissue samples (*n* = 5) were obtained from kidneys that were surgically removed due to renal cell carcinoma (distant nontumorous tissue from the contralateral pole of the kidney) and transplant kidney tissue (*n* = 7) was obtained from explanted renal allografts after transplant failure. These samples are referred to as healthy and transplant kidney samples, respectively. Slices of kidney cortex were chopped into 1-mm cubes with the McIlwain Tissue Chopper (Ted Pella, Redding, CA, USA), transferred to 50 mL tubes and washed with cold PBS until no blood was visibly present and the supernatant was clear. Preheated (37 °C) digestion medium was added, 40 mL per 10 g of tissue (DNAse I type IV (50 KU/mL) (Sigma Aldrich, Zwijndrecht, The Netherlands), collagenase type IV (0.5 mg/mL) (Wortington Biochemical, Lakewood, NJ, USA), BSA (60 mg/mL) (Sigma Aldrich), 20 μL/mL fetal calf serum (FCS, VWR International BV, Amsterdam, The Netherlands), TRIS (0.025 M) (Merck BV, Amsterdam, The Netherlands), penicillin streptomycin (Biochrom GMBH, Berlin, Germany) in HBSS (Westburg BV, Leusden, The Netherlands)), and incubated in a shaker for 20 min at 37 °C. The warm suspension was transferred to a C-tube (Miltenyi, Bergisch Gladbach, Germany) and subjected to the M_spleen_04.01 program on the GentleMacs (Miltenyi). The digestion medium was deactivated with cold PBS and the resulting cell suspension passed through a cell strainer to obtain a single cell suspension, which was subjected to standard density gradient centrifugation according to manufacturer’s protocol (Lymphoprep, Abbott Diagnostics Technologies AS, Oslo, Norway). The isolated mononuclear cells (kidney MNCs) were cryopreserved until the day of analysis in IMDM supplemented with 20% FCS, 0.00036 *v*/*v*% β-mercaptoethanol, 5% DMSO, penicillin and streptomycin.

### 2.4. Flow Cytometry

Measurements were performed on an LSRFortessa flow cytometer (BD Biosciences). For each sample, 2 × 106 PBMCs or 0.5 × 106 to 10 × 106 kidney MNCs were analyzed. The volume of each staining reaction was relative to the number of cells and the antibody concentrations remained constant. Cells were incubated with the surface antibodies (Appendix A) for 30 min at 4 °C in the dark. Dead cells were excluded using the fixable viability dye eFluor455UV (eBioscience Inc., Thermo Fisher Scientific, San Diego, CA, USA). Monoclonal antibodies with intracellular targets (Appendix A) were added after the fixation and permeabilization of the cells using the FoxP3/Transcription Factor Staining Set (eBioscience Inc.). Published methods for flow cytometry and cell sorting for immunological purposes were followed [8]. The gating strategy used for the phenotypic analysis can be found in Appendix A. We have previously shown that none of the markers analyzed were affected by the digestion method used to isolate kidney MNCs [7].

Tissue sample limitations resulted in the exclusion of samples from certain panels. CD3+ cell counts analyzed per sample in healthy PBMCs, healthy kidney MNCs and TX kidney can be found in Appendix A. MNCs Samples were only analyzed when they contained over 50% live cells within the lymphogate, as assessed by the viability dye, and CD69/CD103 based T cell-subsets were only further characterized if their total cell count exceeded 50.

### 2.5. Stimulation Assay

PBMC and kidney MNC were stimulated as previously described [7,9]. PBMCs and kidney MNCs were thawed in the presence of DNAse I (200 KU/mL), washed, and allowed to recover overnight in untreated, round-bottom, 96-well plates (Corning) in culture medium (RPMI supplemented with 10% FCS and penicillin streptomycin) at a concentration of 20 × 106/mL (100 μL/well).

The next morning, phorbol 12-myristate 13-acetate (PMA, 10 ng/mL; Sigma Aldrich) and ionomycin (1 μg/mL; Sigma Aldrich) were added to stimulate the cells. Medium alone was added as the negative control. All incubations were performed in culture medium in the presence of αCD28 (clone 15E8; 2 μg/mL), αCD29 (clone TS 2/16; 1 μg/mL), brefeldin A (10 μg/mL, Invitrogen), and GolgiStop (BD Biosciences) in a final volume of 200 μL for 4 h at 37 °C and 5% CO_2_.

Subsequently, the cells were incubated for 30 min with the surface antibodies (Appendix A). Dead cells were excluded using fixable viability dye eFluor780 (eBioscience Inc., Thermo Fisher Scientific, San Diego, CA, USA). Monoclonal antibodies for intracellular staining (Appendix A) were added after fixation and permeabilization of the cells using the Cytofix/Cytoperm Reagent Set (BD Biosciences). Cells were washed twice and analyzed on an LSRFortessa flow cytometer. The gating strategy used in the functional analysis can be found in Appendix A. To determine the polyfunctionality of each T cell subset, the average number of functions of each population was calculated using the following formula: (([percentage of cells producing 1 cytokine]*1) + ([percentage of cells producing 2 cytokines]*2) + ([percentage of cells producing 3 cytokines]*3) + ([percentage of cells producing 4 cytokines]*4) + ([percentage of cells producing all 5 cytokines]*5))/100.

### 2.6. Data Analysis

Data were analyzed using FlowJo version 10 (FlowJo, Ashland, OR, USA). All graphs and figures were created using Graphpad Prism version 8.00 for Windows (GraphPad Software, La Jolla, CA, USA). The same program was also used for the statistical analyses of the data. The nonparametric Mann–Whitney-U test was used to determine significance of unpaired samples. To compare paired samples, the Wilcoxon signed rank test was used. *p*-values < 0.05 were considered statistically significant.

## 3. Results

### 3.1. Distinct Expression of CD8 and/or CD4 and Markers of Tissue Residency by T Cells in Healthy Kidney Tissue

First, we compared the expression of CD4 and CD8 by CD3-positive T cells in peripheral blood to cells detected in healthy kidney tissue. CD4^−^CD8^+^ (CD8) T cells were significantly more often detected in kidney tissue than in blood ((median) 25% vs. 38%) (Figure 1a). In contrast, CD4^+^CD8^−^ (CD4) T cells were detected in higher frequencies in blood than in kidney tissue ((70% vs. 52%) Figure 1a). Overall, CD4 T cells were seen more frequently in healthy kidney tissue than CD8 T cells (Figure 1b).

Next, we investigated the expression of the tissue-residency markers, CD69 and CD103, in CD8 and CD4-defined T cell subsets. As expected, CD69 and CD103 were virtually not expressed by cells in peripheral blood (Figure 1c). In healthy tissue, the CD69^−^CD103^−^, CD69^+^CD103^−^, CD69^+^CD103^+^, and CD69^−^CD103^+^ phenotypes were expressed by, 46%, 45%, 6.6%, and 1.8% of CD8 T cells, 61%, 38%, 0.44%, and 0.43% of CD4 T cells, respectively (Figure 1c).

In conclusion, CD8, but particularly CD4 T cells were detected in substantial numbers in healthy human kidney tissue. Furthermore, expression of CD69 and CD103, markers of tissue-residency, was detected primarily among T cells found in healthy tissue and not in blood.

### 3.2. Different Expression Patterns of Common Subset-Denominating Markers by T Cells in Healthy Kidney Tissue

CD45RA, a tyrosine phosphatase, CCR7, a chemokine receptor known to mediate T cell trafficking to secondary lymphoid organs, and CD28 and CD27, both costimulatory receptors strongly involved in T cell activation, are all markers traditionally used to identify functional T cell subsets [10,11]. We wanted to investigate to what extent the distribution of these subsets in blood differ from the distribution in kidney tissue. As expected, the largest CD8 T cell subsets detected in the circulation (i.e., those concerning ≥5% of the total population) were those with a CD45RA^+^CCR7^+^CD28^+^CD27^+^ (naive T cells or T_N_, (median) 13%), CD45RA^−^CCR7^+^CD28^+^CD27^+^ (central-memory T cells or T_CM_, 6%), CD45RA^−^CCR7^−^CD28^+^CD27^+^ (T_EM_1, 30%), CD45RA^−^CCR7^−^CD28^+^CD27^−^ (T_EM_2, 10%), CD45RA^−^CCR7^−^CD28^−^CD27^+^ (T_EM_3, 6%), CD45RA^−^CCR7^−^CD28^−^CD27^−^ (T_EM_4, 6%) and CD45RA^+^CCR7^−^CD28^−^CD27^−^ (T_EM_RA, 10%) phenotype. In healthy tissue, almost no (0.6%) CD8 T cells were detected with a T_N_ phenotype and only few (0.8%) displayed a T_CM_ phenotype. In contrast, CD8 T cells detected in healthy kidney tissue were composed of considerable T_EM_1- (22%), T_EM_2- (12%), T_EM_3- (18%), T_EM_4- (23%), and T_EM_RA (11%) T cell populations (Figure 1d and Appendix A).

When looking at CD4 T cells in blood, the largest populations were those expressing a T_N_- (35%), T_CM_- (25%), T_EM_1- (16%), T_EM_2- (9%), and a population with a not otherwise specified CD45RA^+^CCR7^−^CD28^+^CD27^+^ phenotype (0.4%) (Figure 1d and Appendix A). In healthy kidney tissue, few CD4 T cells expressed a T_N_- (3%) or a T_CM_- (3%) phenotype. Instead, a substantial number of cells with a T_EM_1- (22%), T_EM_2- (40%), or T_EM_4- (14%) phenotype were detected in healthy tissue (Figure 1d and Appendix A). 

In conclusion, the distribution of CD45RA/CCR7/CD28/CD27-defined CD8 and CD4 T cell subsets differs substantially between blood and healthy kidney tissue.

### 3.3. CD8 and CD4 T Cells in Healthy Human Kidney Tissue More Often Comprise Actively Cycling Cells

Next, we investigated whether there are differences in the expression of markers typical of a cytotoxic effector-memory profile, between CD8 and CD4 T cell populations in blood and kidney tissue.

First, we looked at the expression of the T-box transcription factors T-bet and eomesodermin (Eomes). These transcriptional regulators are directly involved in generating acute phase effector cells, inducing expression of molecules like interferon-γ and granzyme B, and in the generation of secondary memory responses. As expected, expression of T-bet was frequent among circulatory CD8 T cells (54%), while circulatory CD4 T cells rarely expressed this transcription factor (13%). Interestingly, for CD4 T cells, T-bet expression was significantly higher among T cells detected in healthy kidney tissue than in the circulation (53% vs 13%). Eomes expression was also detected frequently among circulatory CD8 T cells (59%), while again being rare among circulatory CD4 T cells (15%).

However, Eomes expression was more frequent among CD4 T cells in kidney tissue (32%) than in blood (Figure 2a,b).

Next, we investigated expression of granzyme B, a serine protease that is known to mediate apoptosis after injection into the cytoplasm by cytotoxic T cells. In line with previous reports, T-bet and granzyme B expression were detected in substantial numbers in circulating CD8 T cells (28%). However, its expression was rare among circulating CD4 T cells (1%). For CD8 T cells, granzyme B expression was similar in kidney tissue and circulating populations (28% vs. 23%) (Figure 2c). Interestingly, the higher T-bet and Eomes expression frequencies detected in kidney CD4 T cells did not correspond with a higher expression frequency of granzyme B in this compartment (5%). We then looked at expression of KLRG1, a coinhibitory receptor known to be predominantly expressed by cytotoxic acute phase effector cells and effector-memory cells. KLRG1 was expressed frequently by circulating CD8 T cells (54%), but not by CD4 T cells (10%). No differences were found between anatomical compartments in KLRG1 expression for CD8 or CD4 T cells. Lastly, we investigated the expression of Ki-67, a marker denoting actively cycling cells. In line with previous observations, Ki-67 expression among circulating CD8 and CD4 T cells was rare (1% and 2%, respectively). Interestingly, among CD8 T cells, but particularly among CD4 T cells, Ki-67 expression was significantly more often detected in T cells in kidney tissue (3% and 11%, respectively) (Figure 2e).

In conclusion, CD4 T cells detected in healthy human kidney tissue, but not CD8 T cells in this compartment, more frequently expressed T-bet and Eomes. However, this did not translate into a more frequent expression of granzyme B by kidney CD4 T cells. Moreover, although overall numbers of cycling cells were low, kidney tissue contained significantly more actively cycling CD8 and CD4 T cells than the circulation.

### 3.4. CD8 and CD4 T Cells in Healthy Kidney Tissue are Nearly All CXCR3-Positive

We then looked for differences in expression of other markers, typical in circulating populations of not immediately cytotoxic memory cells, as shown previously for circulating T cell populations. We first looked at the expression of IL-7Rα, expressed primarily on naïve and early-differentiated memory cells in the circulation, where it is involved in sustaining homeostatic proliferation [10,12,13]. In peripheral blood, IL-7Rα was expressed frequently by CD8 T cells (84%) and CD4 T cells (98%). CD8 and CD4 T cells in healthy kidney tissue expressed this marker significantly less often than circulating T cells (71% and 76%, respectively) (Figure 2f).

Next, we investigated the expression of various other chemokine receptors. Interestingly, CXCR3, a chemokine receptor involved in T cell trafficking to inflamed tissues [14,15,16,17], was expressed by exceedingly large numbers of CD8 and CD4 T cells in kidney tissue (99% and 91%, respectively), and significantly less often by circulating T cells (58% and 26%, respectively) (Figure 2g). Finally, we determined the expression of CXCR6, known to be implicated in forming T_RM_ populations [18,19,20,21,22]. While CXCR6 was expressed by only a small population of circulating CD8 T cells (16%), and virtually not by circulating CD4 T cells, a significantly larger proportion of CD8 and CD4 T cells in kidney tissue expressed this molecule (40% and 40%, respectively) (Figure 2h).

In conclusion, markers typically associated with a not immediately differentiated state in the circulation, like IL-7Rα, CCR7, CD28, and CD27, were significantly less often expressed by kidney CD8 and T cells when compared to those in blood. In contrast, kidney T cells more often expressed CXCR6 and almost always expressed CXCR3.

### 3.5. Distinctions in Phenotype between Healthy Kidney and Allograft T Cells Are Minimal

Next, we wanted to investigate whether these findings remained standing in tissue from transplanted kidney allografts that were obtained for various clinical indications (Table 1).

With regard to the numbers of CD8 and CD4 T cells in healthy and allograft tissue, no differences were seen (Figure 3a). However, we noticed a nonsignificant trend towards more CD4 T cells, and less CD8 T cells in healthy tissue than in allograft tissue (Figure 3b). Additionally, when looking at CD69/CD103 expression and CD45RA/CCR7/CD28/CD27-defined subsets, no differences in distribution of CD69/CD103- or CD45RA/CCR7/CD28/CD27-defined subsets were noted, regardless of CD8/CD4 phenotype (Figure 3d and Appendix A). When examining expression frequencies of the markers more typical for cytotoxic circulating T cells, we found no differences in expression of T-bet, KLRG1 or granzyme B between healthy kidney or allograft T cells (Figure 3a,g,h). We did find that Eomes expression was seen less frequently in allograft tissue CD4 T cells, than in healthy tissue T cells (Figure 3f). No differences in the expression of the other markers were observed. (Figure 3i–l and Appendix A).

In conclusion, apart from a lower expression frequency of Eomes among allograft CD4 T cells, cell populations did not differ between study populations.

### 3.6. CD69^−^CD103^−^ T Cells in Kidney Tissue Are Different from Circulating T Cells

Given the similar phenotypic makeup of CD8 and CD4 T cells found in healthy kidney-tissue and in kidney allograft tissue, we pooled data from both study groups to further examine characteristics of these cells according to CD69 and CD103 expression. Since CD69^−^CD103^+^ cells were always detected in very low frequencies (<50 events), we excluded these cells from further analyses.

First, we investigated differences in these subsets according to CD69/CD103 coexpression in kidney T cell populations (Figure 4 and Appendix A). Interestingly, for both CD8 and CD4 T cells, substantial differences were noted in the distribution of CD45RA/CCR7/CD28/CD27-defined subsets among CD69^−^CD103^−^, CD69^+^CD103^−^, and CD69^+^CD103^+^ subsets. For CD8 T cells, CD69^−^CD103^−^ cells contained a sizeable population of T_EM_RA cells (21%) that was much smaller among the CD69+CD103- and CD69^+^CD103^+^ subsets (5.8% and 2.2%, respectively). Instead, these latter populations contained much larger T_EM_3 (13%, 30%, and 32%, respectively) and T_EM_4 (8%, 19%, and 33%, respectively) subpopulations. For CD4 T cells, few T_EM_RA cells were seen among CD69^−^CD103^−^, CD69^+^CD103^−^, and CD69^+^CD103^+^ subsets (2.6%, 0%, and 1.7%, respectively). Instead, these cells held large subpopulations of T_EM_1 (26%, 24%, and 10%, respectively) and T_EM_2 cells (30%, 46%, and 27%, respectively). Among CD69^+^CD103^−^ and particularly among the CD69^+^CD103^+^ cells, a much larger T_EM_4 subpopulation was seen (6%, 11% and 34%, respectively).

Given these distinctions, and the previous observation that kidney T cell populations barely comprised T_N_ and T_CM_ cells, we further investigated the CD69^−^CD103^−^ cells found in kidney tissue in order to determine to what extent these cells were not just circulating cells passing through the kidney circulation. In order to do so we compared these cells to circulating cells. CD69^−^CD103^−^ cells in kidney tissue differed significantly from those in the circulation: T-bet, Ki-67 and CXCR3 were expressed more often by kidney T cells, regardless of CD8/CD4 phenotype (Figure 5a–c). In contrast, IL-7Rα, CCR7, CD28, and CD27 were all expressed significantly less frequently by CD69^−^CD103^−^ kidney T cells, regardless of CD8/CD4 phenotype (Figure 5d–g). CD69^−^CD103^−^ CD4 T cells in kidney tissue expressed KLRG1 significantly more often than in blood and CD69-CD103- CD8 T cells in kidney tissue expressed Eomes significantly more often than in blood.

Next, we wanted to see how CD69 and CD103 coexpression affected markers of functional potential. First, we looked at markers associated with cytotoxic potential. Although differences were noted between individual subpopulations, no consequent patterns were seen for T-bet and Eomes in CD8 T cells when CD69 or CD103 were coexpressed (Figure 6a,b). In contrast, in CD4 T cells these transcription factors both increased together with CD69 and CD103 coexpression (Figure 6a,b). When looking at KLRG1, we noticed that for CD8 T cells, KLRG1 expression declined together with CD69/CD103 coexpression. For CD4 T cells a similar trend was seen. However, here only CD69^−^CD103^−^ cells expressed KLRG1 at a higher frequency than CD69^+^CD103^−^ and CD69^+^CD103^+^ cells (Figure 6c). Expression frequencies of granzyme B also declined together with markers of tissue-residency, with its expression frequency being lowest amongst CD69^+^CD103^+^ cells, within both CD8 and CD4 T cells (Figure 6d). Expression of Ki-67, which was higher amongst kidney T cells when compared to T cells in blood, did not differ according to CD69 or CD103 coexpression (Figure 6e).

We then investigated markers normally associated with a noncytotoxic memory profile according to CD69/CD103 coexpression. Expression of IL-7Rα, CCR7, and CD28 was highest within the CD69^−^CD103^−^ populations, with a trend in declining expression along with coexpression of CD69/CD103 (Figure 6f–h). We then looked at expression of CXCR3 and CXCR6. Both chemokine receptors were most often expressed by CD69^+^CD103^+^ cells, regardless of CD8/CD4 phenotype and showed an upward trend with coexpression of CD69/CD103. Only for CXCR6 this trend was statistically significant, affecting both CD8 and CD4 T cells (Figure 6j,k).

In conclusion, CD69^−^CD103^−^ cells in kidney tissue differ substantially from cells in the circulation and may concern another T_RM_ population not marked by CD69 or CD103 expression. The only marker that differed consistently, regardless of CD8/CD4 phenotype, in terms of CD69 and CD103 coexpression, was CXCR6.

### 3.7. T Cells in Kidney Tissue More Often Produce Interferon-γ than T Cells in Blood

Finally, we investigated cytokine production capacity by CD8/CD4-defined T cell populations in kidney tissue. First, we compared production of interferon-γ (IFNγ), tumor necrosis factor α (TNFα), interleukin-2 (IL-2), granulocyte-macrophage colony stimulating factor (GM-CSF), and IL-17 by stimulated CD8/CD4-defined T cell populations found in blood to those in healthy kidney tissue (Figure 7a–e). Interestingly, only IFNγ was consistently expressed by a greater proportion of T cells in healthy kidney tissue, when compared to T cells in blood, regardless of CD8/CD4 phenotype (Figure 7a). Differences were also noticed on an individual subset level. Specifically, TNFα and GM-CSF were more frequently produced by CD4 T cells in kidney tissue (Figure 7b,c). With regard to polyfunctionality (i.e., the number of simultaneously produced cytokines), CD4 T cells in kidney tissue were found to be more polyfunctional than CD4 cells in blood (Figure 7f).

Next, we looked at differences in cytokine production capacity between stimulated T cells from healthy and allograft tissue. No differences in production of individual cytokines, or in number of simultaneously produced cytokines were detected between T cells from healthy or allograft kidney tissue (Appendix A).

Since CD69 is expressed upon stimulation of T cells, we only studied the difference between CD103-negative and CD103-expressing CD4 and CD8 T cells (Figure 7g–l). When comparing CD4 and CD8 T cells expressing CD103 with cells not expressing CD103 in kidney tissue, we noticed that CD8 and CD4 CD103^+^ cells expressed more IFNγ than CD103^−^ cells (Figure 7g). Additionally, although IL-17A-producing cells were low among CD103^+^ CD8 and CD4 T cells in kidney tissue, they were consistently found more frequently in this population compared to the CD103^−^ population.

In conclusion, more T cells in kidney tissue produced IFNγ when compared to circulating T cells, regardless of CD8/CD4 phenotype. Indeed, the highest percentage of IFNγ-producing cells was found among the CD103-expressing CD4 and CD8 kidney-derived T cells. In addition, T cells in kidney allografts did not differ from those in healthy kidney tissue with respect to these parameters.

## 4. Discussion

In the current study we addressed the diversity of T cell subsets, according to expression of CD8 and CD4, and the markers of tissue-residency, CD69 and CD103, in human kidneys and compared to these to blood.

Substantial numbers of CD8 and CD4 T cells were indeed detected in kidney tissue. Furthermore, these T cells in kidneys contained considerable numbers of CD69^+^CD103^−^ cells. CD69+CD103+ cells were also detected, but predominantly among the CD8 T cells and were virtually nonexistent among CD4 T cells. A similar finding concerned lung T_RM_ previously [23]. As expected, such CD69 and CD103-expressing populations were instead near absent from the circulation. A striking difference when comparing T cells in kidney with those in blood, concerned a distinct distribution of the traditional CD45RA/CCR7/CD28/CD27-defined subsets. Interestingly, expression of CD28 was more often not expressed when T cells were coexpressing CD69 alone or in combination with CD103. Theoretically, this should render these populations less susceptible to costimulatory signals by CD80 and CD86. Other differences concerned a higher expression frequency of CXCR3, CXCR6, and Ki-67. Additionally, kidney T cells more often contained IFNγ-producing cells and were frequently polyfunctional. Indeed, within the kidney, CD103+ T_RM_s contained more IFNγ-producing cells than their CD103-negative counterparts. In addition, CD103^+^ T cells more often produced IL-17, although this concerned a modest population.

With regard to the more frequent expression of CXCR3 by CD8 and CD4 kidney T cells, the CXCR3/CXCL10 axis was previously shown to be involved in the shaping of T_RM_ populations in skin, liver, and lymphoid tissue [24,25,26,27,28]. Additionally, it is involved in the trafficking to stressed epithelium and inflammation in a general sense [14,15,16,17]. Given the high frequencies of CXCR3 among these T cells in healthy kidney tissue, CXCR3 appears to be implicated in T_RM_ homeostasis in kidney tissue as well and may not necessarily involve inflammation or danger signals to do so. With regard to CXCR6, the CXCR6/CXCL16 axis has indeed already been shown to be essential for trafficking of T_RM_ to different compartments like the lungs, skin, liver, and brain, and may concern a universal mechanism essential for guiding and retaining T_RM_ to and in tissues, also concerning T_RM_ in human kidney tissue [18,19,20,21,22].

Farber and coworkers have extensively characterized and compared CD8 and CD4 T_RM_ populations residing in different human tissue types derived from organ donors. They described how TRM populations in lung, spleen, small intestines, and various secondary lymphoid tissues differ with regard to transcriptome, but also to expression of various proteins, including cytokines, investigated in the current study [29,30,31]. However, these investigations did not comprise TRM populations in the human kidney, and data on this subject is scarce. We previously showed that CD8 T cells targeting polyomavirus BK VP1 and LTAG proteins, deriving from a virus that has a strong tropism for renal epithelial cells, are detectable in allograft kidneys. These virus-specific T cells expressed CD69 and CD103 in higher numbers than their counterparts found in the circulation of the same patients. Furthermore, these T cells generally expressed a CD45RA^−^CD27^−^ granzyme B-negative phenotype [5]. Later, others have also reported on T_RM_ populations in human kidney tissue obtained from transplant nephrectomies. In this publication, the authors report that T cells in these tissues mainly concerned CD8 T cells [32]. We instead found that CD4 T cells, in fact, comprised the largest T cell subset. However, we also found that the balance shifted, although not significantly, towards greater numbers of CD8 T cells in allograft tissue when compared to healthy kidney tissue. As such, this discrepancy might be explained by a different study group, and it may be that in health, the immunological defense is more focused on extracellular threats, like bacteria, versus more intracellular threats, like viruses and cancer in the immunocompromised hosts. We did not find other differences in the markers studied, between healthy kidney and allograft T cells. However, further research, using techniques with a broader view on transcriptome, like RNA sequencing, or proteome, like mass spectrometry, should be employed to establish whether this is indeed the case.

In addition, de Leur et al. describe how T_RM_ in kidney frequently expressed granzyme B, while we detected only low amounts of granzyme B in both healthy and allograft kidney tissue [32]. However, we did detect substantial T-bet and Eomes-expressing T cells in kidney tissue, which in circulating T cell populations, is normally indeed associated with a higher expression of granzyme B [10]. While the latter may not be surprising, given the fact that granzyme B expression is also directly induced by T-bet [33,34], this association was not apparent among kidney T cells in the current study. As we and others have shown that also human lung-, brain-, and skin-derived T_RM_ [25,26,35,36], as well as kidney-derived tissue resident MAIT cells [6,7] are low in their granzyme B content, while often expressing substantial amounts of T-bet and Eomes, this might be a common trait of (human) T_RM_. Furthermore, in lung T_RM_ granzyme B expression was seen to be rapidly upregulated upon stimulation [35], suggesting that this might also occur in other T_RM_. Here, the authors hypothesized that such ‘hidden cytotoxicity’ serves to protect the structural integrity of tissues from damage by cytotoxic T cells.

In conclusion, we show that T cells in kidney tissue are present in substantial numbers and comprise populations that frequently express CD69^+^ and CD103^+^, molecules by which T cells adhere to, and persist in, tissues. Nevertheless, the CD69^−^CD103^−^ T cells in kidney tissue appear to be distinct from circulating T cells and may concern another distinct T_RM_ population, possibly persisting in kidneys by other mechanisms yet to be unveiled. Another option may be that these CD69^−^CD103^−^ T cells concern a population that have only recently egressed from the circulation, and have only transiently acquired new phenotypic traits due to an interaction with the kidney microenvironment. Further investigations into this population are needed to elucidate these possibilities. Furthermore, CD8 and CD4 kidney T cells, were often actively cycling and more frequently expressed CXCR6. Additionally, CD8 and CD4 populations displayed a particularly high expression of CXCR3, strongly implicating these chemokine receptors, as well as CXCR6 in T_RM_ accrual and persistence in human kidneys. Therefore, further investigations are needed to enhance our image of T_RM_ populations in the human kidney. Nevertheless, the findings presented here form a solid first step towards more detailed characterization of T cell populations in kidney tissue and their role in health and disease.

## Figures and Tables

**Figure 1 cells-10-00288-f001:**
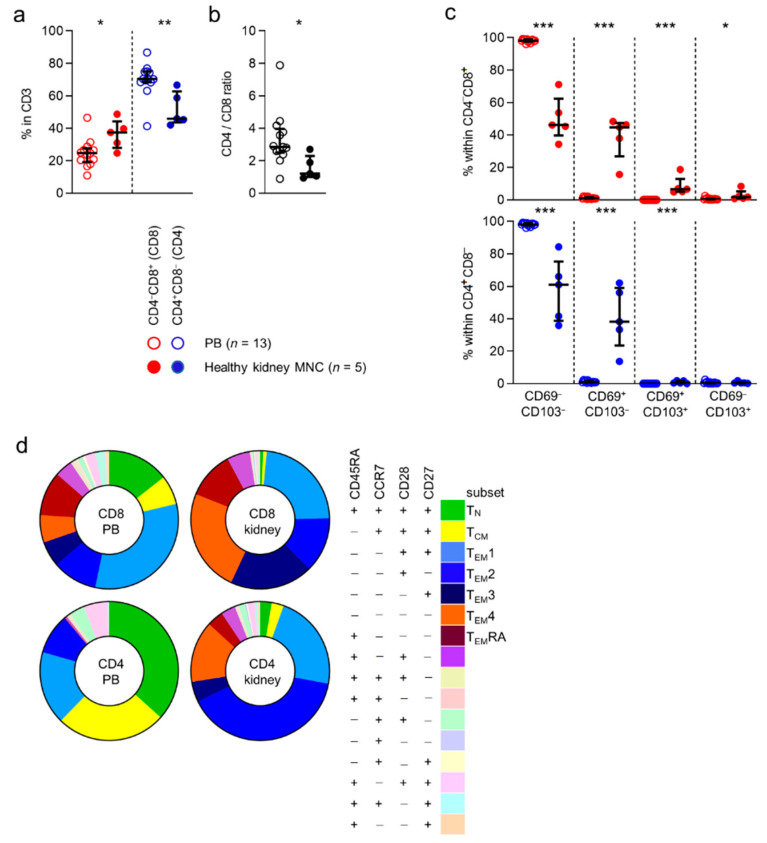
Kidney-derived T cells differ in CD4/CD8 distribution, contain TrM and have a different memory-profile than PB T cells. (**a**) Percentage of CD4^−^CD8^+^ (CD8) and CD4^+^CD8^−^ (CD4) T cells within CD3^+^CD45^+^ single-cell live lymphocytes in peripheral blood (PB) and healthy kidney MNC (median with IQR in black). (**b**) CD4/CD8 ratio within PB and healthy kidney-derived T cells. (**c**) Distribution of CD69/CD103 based subsets within CD8 and CD4 T cells (median with IQR in black) derived from PB and healthy kidney. (**d**) Expression patterns of CD27, CD45RA, CCR7 and CD28 within CD8 and CD4 T cells from PB and healthy kidney (median). Mann-Whitney U-test was used for statistical comparison. Only significant p-values are displayed: * *p* < 0.05, ** *p* ≤ 0.01, *** *p* ≤ 0.001.

**Figure 2 cells-10-00288-f002:**
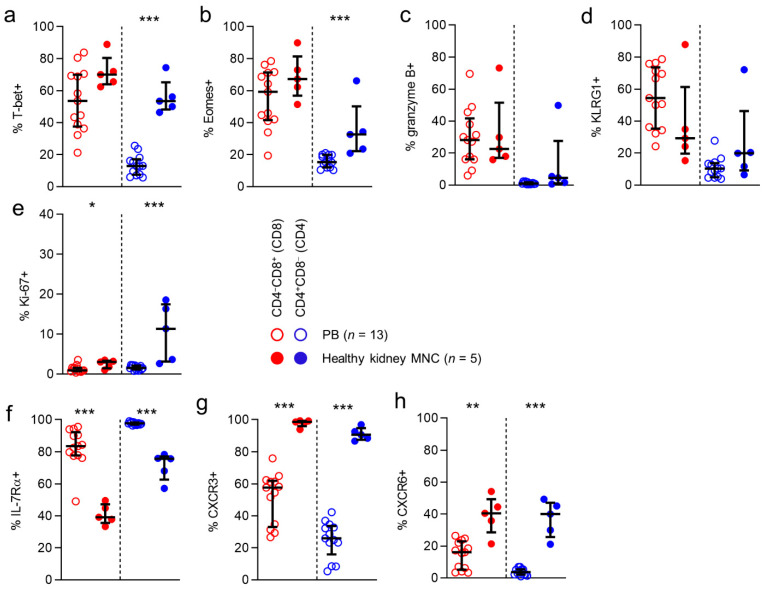
CD8 and CD4 T cells in kidney tissue are nearly all CXCR3-positive.(**a**–**h**) Comparison of expression of T-bet (**a**), Eomes (**b**), granzyme B (**c**) KLRG1 (**d**), ki-67 (**e**), IL-7Rα (**f**), CXCR3 (**g**) and CXCR6 (**h**) in CD8 (red) and CD4 (blue) T cells derived from peripheral blood (PB) and healthy kidney. (median with IQR in black). Mann-Whitney U-test was used for statistical comparison. Only significant p-values are displayed: * *p* < 0.05, ** *p* ≤ 0.01, *** *p* ≤ 0.001.

**Figure 3 cells-10-00288-f003:**
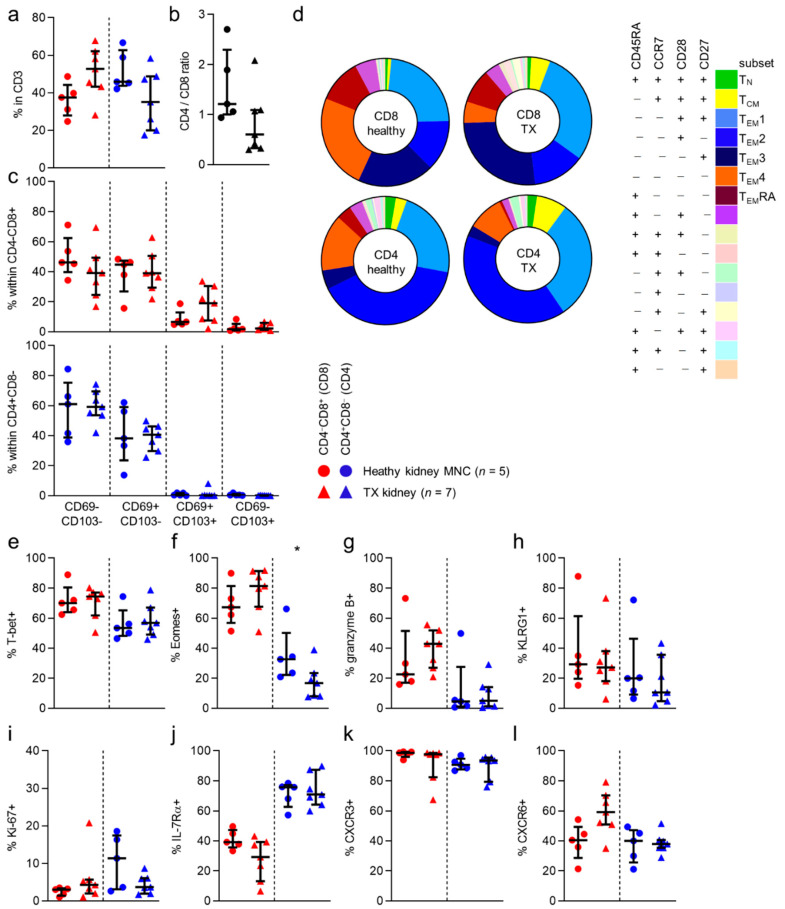
No differences in phenotype between healthy kidney and allograft T cells.(**a**) Percentage of CD4^−^CD8^+^ (CD8) and CD4^+^CD8^−^ (CD4) T cells within CD3^+^CD45^+^ single-cell live lymphocytes from healthy kidney compared to transplanted (TX) kidney (median with IQR in black). (**b**) CD4/CD8 ratio within PB and healthy kidney-derived T cells. (**c**) Distribution of CD69/CD103 based subsets within CD8 and CD4 T cells (median with IQR in black) derived from healthy and TX kidney (median with IQR in black). (**d**) Expression patterns of CD27, CD45RA, CCR7 and CD28 within CD8 and CD4 T cells from healthy kidney and TX (transplanted) kidney (median). (**e**–**l**) Comparison of expression of T-bet (**e**), Eomes (**f**), granzyme B (**g**), KLRG1 (**h**), ki-67 (**i**), IL-7Rα (**j**), CXCR3 (**k**) and CXCR6 (**l**) in CD8 (red) and CD4 (blue) T cells derived from healthy kidney and TX kidney (median with IQR in black). Mann-Whitney U-test was used for statistical comparison. Only significant *p*-values are displayed: * *p* < 0.05.

**Figure 4 cells-10-00288-f004:**
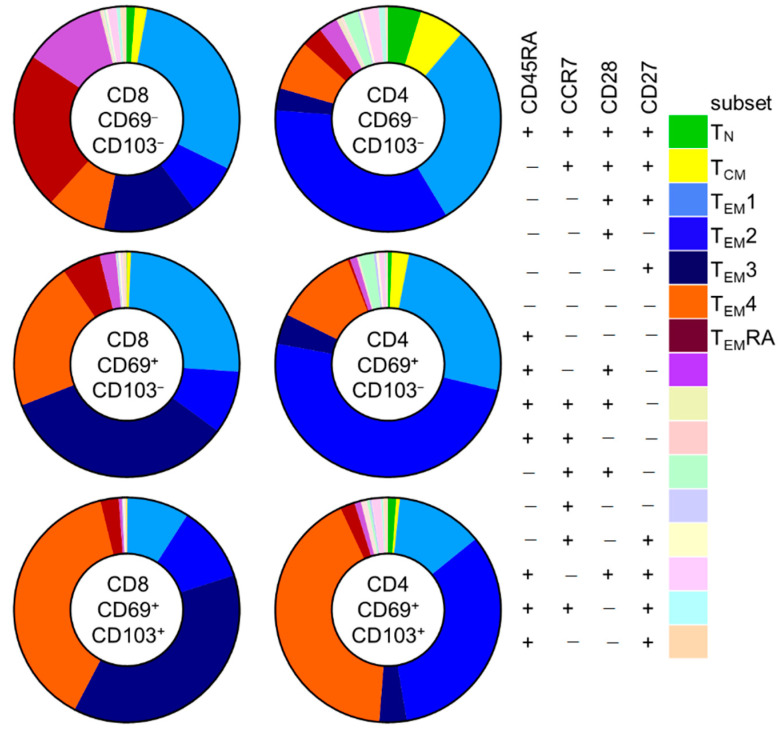
Memory-subset composition within CD4 and CD8 T cells from kidney changes upon CD69/CD103 co-expression. Expression patterns of CD45RA, CCR7, CD28 and CD27 within in CD69^−^CD103^−^, CD69^+^CD103^−^ and CD69^+^CD103^+^ subsets within CD8 and CD4 T cells from kidney (median).

**Figure 5 cells-10-00288-f005:**
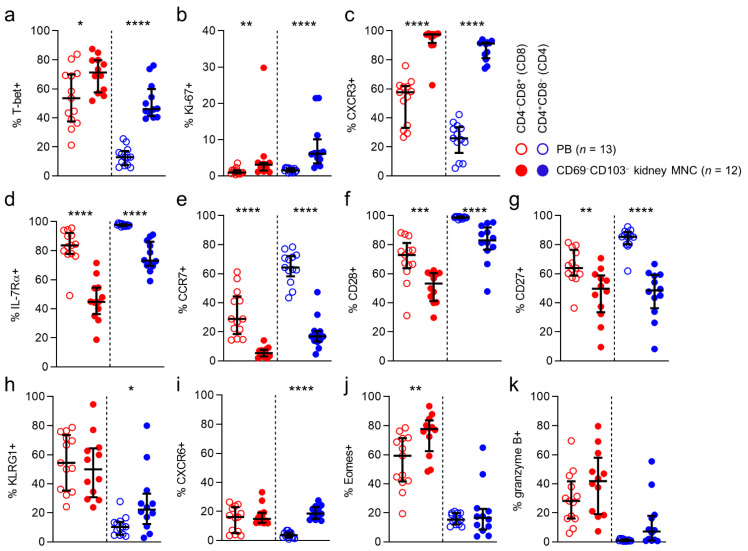
**CD69^−^CD103^−^ T cells in kidney tissue differ substantially from cells in the circulation.** Comparison of expression of T-bet (**a**), ki-67 (**b**), CXCR3 (**c**), IL-7Rα (**d**), CCR7 (**e**), CD28 (**f**), CD27 (**g**), KLRG1 (**h**), CXCR6 (**i**), Eomes (**j**) and granzyme B (**k**) in CD8 (red) and CD4 (blue) in peripheral blood (PB)-derived T cells and kidney-derived CD69^−^CD103^−^ T cells (median with IQR in black). Mann-Whitney U-test was used for statistical comparison. Only significant p-values are displayed: * *p* < 0.05, ** *p* ≤ 0.01, *** *p* ≤ 0.001, **** *p* ≤ 0.0001.

**Figure 6 cells-10-00288-f006:**
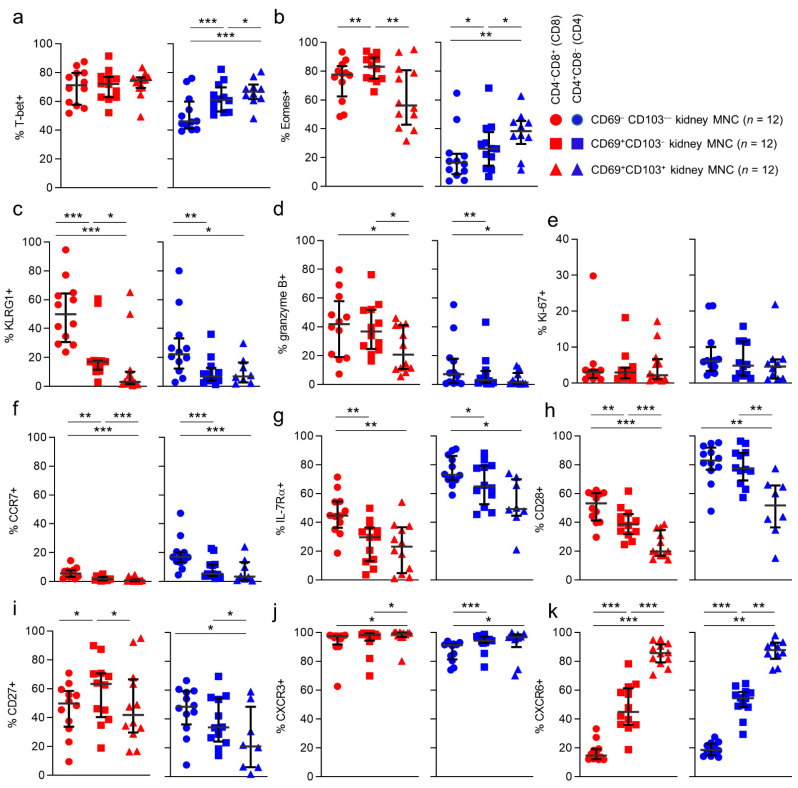
**CXCR6 is consistently more often expressed by cells coexpressing CD69 and CD103.** (**a**–**k**) Comparison of expression of T-bet (**a**), Eomes (**b**), KLRG1 (**c**), granzyme b (**d**), ki-67 (**e**), CCR7 (**f**), IL-7Rα (**g**), CD28 (**h**), CD27 (**i**), CXCR3 (**j**) and CXCR6 (**k**) in CD69^−^CD103^−^, CD69^+^CD103^−^ and CD69^+^CD103^+^ subsets within CD8 (red) and CD4 (blue) T cells derived from kidney (median with IQR in black). Wilcoxon signed rank test was used for statistical comparison. Only significant p-values are displayed: * *p* < 0.05, ** *p* ≤ 0.01, *** *p* ≤ 0.001.

**Figure 7 cells-10-00288-f007:**
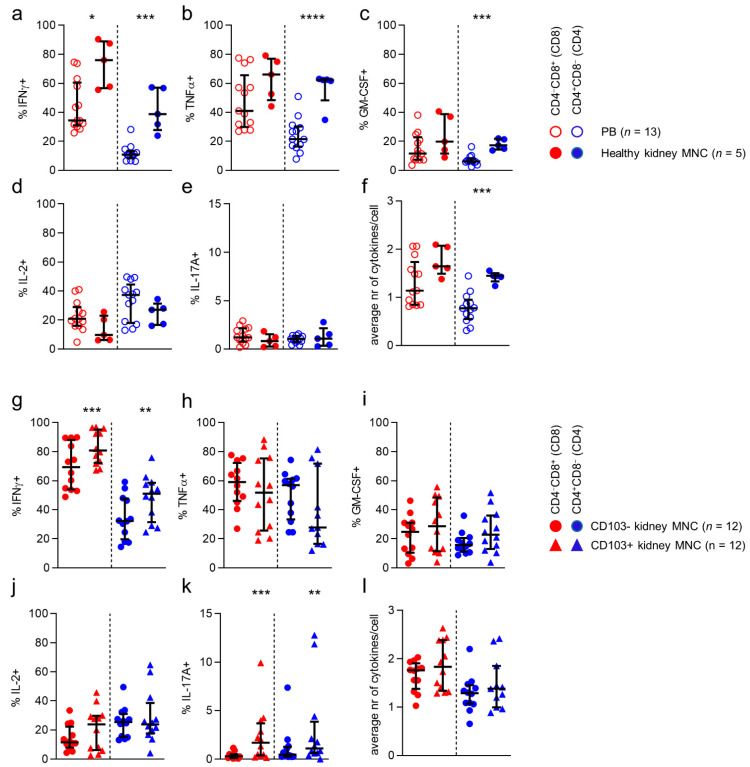
**CD4 T cells in kidney tissue are more often polyfunctional than their counterparts in blood.** (**a**–**e**) Percentage of IFNγ-(**a**), TNFα-(**b**), GM-CSF-(**c**), IL-2-(**d**) and IL-17A-(**e**) producing CD8 (red) and CD4 (blue) T cells derived from peripheral blood (PB) and healthy kidney after 4 hr stimulation with PMA and ionomycin (median with IQR in black). (**f**) The average number of cytokines produced by CD8 and CD4 T cells derived from PB and healthy kidney (median with IQR in black). (**g**–**k**) Percentage of IFNγ-(**a**), TNFα-(**b**), GM-CSF-(**i**), IL-2-(**j**) and IL-17A-(**k**) producing CD103^−^ and CD103^+^ CD8 (red) and CD4 (blue) T cells derived from kidney after 4 hr stimulation with PMA and ionomycin (median with IQR in black). (**l**) The average number of cytokines produced by CD103- and CD103+ CD8 and CD4 T cells derived from kidney (median with IQR in black). Statistical comparisons used: (**a**–**d**) Mann-Whitney U-test (**e**,**f**) Wilcoxon signed rank test. Only significant p-values are displayed: * *p* < 0.05, ** *p* ≤ 0.01, *** *p* ≤ 0.001.

**Table 1 cells-10-00288-t001:** Demographics.

	Healthy Kidneys	Kidney Transplants	Healthy Donors
	*n* = 5	*n* = 7	*n* = 13
Sex, men (%)	60%	57%	62%
Age in years, median (range)	71 (69–78)	55 (26–71)	57 (33–70)
Diabetes mellitus (%)	Unknown	14%	0%
Original renal disease of the allograft recipient (%)	N/A		N/A
- FSGS		2	
- Renovascular		2	
- IgAN		1	
- ADPKD		1	
- Idiopathic membranous glomerulopathy		1	
- Unknown		-	
Cause of transplant failure as reported in the medical file (%)	N/A		N/A
- Rejection		5	
- ATN		1	
- BK		1	
Months since renal transplantation, median (range)	N/A	14 (0.3–60)	N/A
Number of immunosuppressive agents at time of sample collection, median (range)	N/A	2 (1–4)	N/A
Donor type (living/NHB/HB)	N/A	3;4;0	N/A

N/A: not applicable; FSGS: focal segmental glomerulosclerosis; IgAN: IgA nephropathy; ADPKD: autosomal dominant polycystic kidney disease; ATN: acute tubular necrosis; BK: polyomavirus BK-induced nephropathy.

## Data Availability

All data is contained within the article or supplementary material. Raw data are available on request.

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
