# Peer review of "CD8 and CD4 T Cell Populations in Human Kidneys"

_cells, 2021, doi:10.3390/cells10020288_

Round 1
Reviewer 1 Report
In the current manuscript the authors performed a detailed phenotyping of human CD4 and CD8 T cells in kidney tissues and peripheral blood. Albeit the study is extremely descriptive it is of importance to characterize tissue lymphocytes via complex flow cytometry. The data is represented in a clear outline.
Major point:
The authors should not only focus on analyzing CD4 and CD8 T cell populations and include results of CD4/CD8 double negative CD3+ T cells throughout the manuscript. This subset represents the gamma delta T cell lineage, which have been shown to often reside in tissues. It would be of interest to the reader to also characterize gamma delta T cells in such valid cohort of biosamples. Depending on the results the authors may change the manuscript title: "Flow cytometric characterization of T cell populations in human kidneys"
Reviewer 2 Report
In the present study the Authors describe the phenotypic characteristics of T-cells present in renal tissues harvested from kidneys either removed for the presence of a cancer (considered as a normal control tissues) or from kidney grafts removed after transplant failure, in comparison with with the T-cell phenotype in the peripheral blood.
By checking for a long array of markers specific for a number of T-cell function (trafficking, co-stimulation, central-memory, effector-memory, etc), the Authors conclude that the characteristics of T-cell phenotype present in renal tissue is quite different from that one of circulating T-cells, without any major significant difference between T-cell present in “normal” renal tissue and that one observed in the explanted renal grafts.
I think that this big effort gave rise to results which return very limited information.
I will limit myself to describe which, in my opinion, are the main limitations of the present study:
- The comparison between the phenotype of T-cell present in the renal tissue and that one of circulating T-cells is meaningless, since it is well and long known that T-cells resident in the tissues have different characteristics from circulating cells. It could have had more informative value to know whether T-cells in renal tissue present different characteristics as compared with T-cell resident in other tissues ( cutis,liver, lung, etc..)
- Furthermore, it is quite unusual not finding any difference between the T-cells infiltrating a graft as compared with a normal tissue. It is quite possible that the lack of found differences is at least in part due to methodological issues or to the low number of samples. In addition, one should consider that an explanted graft is not the ideal tissue for sampling the phenotype of infiltrating T-cells, since it is expected that an explanted graft can present either an end-stage kidney disease and/or the results of many immunesuppressive drugs, used in the long run or acutely at high dose in the extreme attempt for avoiding the graft loss. In other word, it is not the ideal renal tissue sample for an allograft; renal biopsy samples might have been more appropriate.
- Results section: it is not clear to which denominator the percentage of the different T-cell phenotypes makes reference: is it the total number of mononuclear cells observed in the tissue? If so, which the dimension of the considered histological area?
- Results section: The Authors state that undefined functional CD8 T-cells (CD45RA+CCR7-CD28+CD27-) was found higher in healthy tissue than in peripheral blood, reporting values of 5 and 4 % , respectively: is this difference significant? I don’t think so!
- Results section: when the Authors deal with T-bet they first report that 15 % of circulatory CD4 T-cells express this marker, but three lines after they report 13 %.
- In all the figures it is not easily understandable which are the panels reporting on CD8 T-cells and those reporting on CD4 T-cells. The Authors should specify that left panels refer to the CD8 T-cells and the right panels to the CD4 T-cells, or even better, the Authors should use different symbols (e.g. squares for CD8 and circles for CD4).
- An acronym list could be useful
- The English language could be improved
- The page number is the same on each page: 9!
Reviewer 3 Report
In this research manuscript, van der Putten et al sampled healthy and allograft transplanted renal tissue for a comprehensive phenotypic analysis of CD4 and CD8 T cell populations. Directly comparisons were made with T cell populations in peripheral blood. The authors show that both CD4 and CD8 T cell populations are found in kidney tissues, and that their phenotype are distinct from those in blood, namely expression of CD69, CD103, CXCR3 and CXCR6. Renal T cells demonstrated a higher capacity to produce inflammatory cytokines and displayed signs of recent cell proliferation. No significant differences in T cell phenotypes between healthy and allograft transplanted renal tissue were observed, however, CD69 and CD103 did differentiate subsets of T cells that displayed altered expression of various transcriptional factors, cytokine receptors and chemokine receptors. Finally, they show that CD69-CD103- T cells displayed features of TRM in kidneys, suggesting a possible unique subset in this organ. The research is conducted very well and the manuscript is well written. This works contributes to the important initial characterization of T cell subsets in the kidneys, where tissue residency by T cells is not well established compared to other organ systems, and further extends the author’s previous reports on MAIT cells in kidney tissues. However, several clarifications are requested, and a greater discussion on several observations is needed before publication. My comments are below:
1. In figure 2e, the % Ki67+ in CD4 and CD8 T cells are shown. While increased percentages of CD4 T cells in the kidney are observed, increase in CD8 T cells is very modest at best, even if statistically significant. The conclusion state in line 251 needs to be tempered to reflect this. Can the authors offer insights on what can be driving increased proliferation in CD4 T cells in this tissue compartment?
2. There is no explanation offered for the increased expression of IL-7Ra in resident CD8 T cells but lower in CD4 T cells. Also with CD28 reduction in resident CD4 and not CD8 T cells, the statement in line 394: “…making these populations less susceptible to co-stimulatory signals by CD80 and CD86, and CD70, respectively.” is not explored further. Is this known to be important in the generation/maintenance of the TRM phenotype?
3. Figure 7K: the y-axis needs to be adjusted so that data points and mean values are visible. Though indicated as statistically different, changes in IL-17A expression between the two populations is minor. Conclusions need to be tempered to reflect this modest change in cytokine expression.
4. Phenotypic analysis of CD69-CD103- T cells in periphery vs kidney is interesting. While the authors state that this may represent a new type of TRM, is it possible that these are peripheral blood-derived T cells that have transiently up-regulated (or down-regulated) markers based on the kidney tissue microenvironment? Can the authors elaborate on how the kidney tissue environment impacts T cell phenotypes and/or function?
5. Phenotypic characterization shown in Figure 4: If little CD69+CD103+ CD4 T cells were observed in renal tissues, are there sufficient numbers of these cells analyzed to draw meaningful conclusions on their memory subset composition? What absolute CD69+CD103+ T cell numbers were recovered during acquisition?
Round 2
Reviewer 2 Report
I acknowledge that the authors have different ideas from mine, but the reasons they pose are not enough for me to change my opinion on what has been said